# Methionine Promotes the Growth and Yield of Wheat under Water Deficit Conditions by Regulating the Antioxidant Enzymes, Reactive Oxygen Species, and Ions

**DOI:** 10.3390/life12070969

**Published:** 2022-06-28

**Authors:** Muhammad Faisal Maqsood, Muhammad Shahbaz, Saba Kanwal, Muhammad Kaleem, Syed Mohsan Raza Shah, Muhammad Luqman, Iqra Iftikhar, Usman Zulfiqar, Arneeb Tariq, Shahzad Amir Naveed, Naila Inayat, Atta Mohi Ud Din, Muhammad Uzair, Muhammad Ramzan Khan, Fozia Farhat

**Affiliations:** 1Department of Botany, University of Agriculture, Faisalabad 38000, Pakistan; faisalmuhammaad864@gmail.com (M.F.M.); shahbazmuaf@yahoo.com (M.S.); sabamehar625@gmail.com (S.K.); kaleemakmal5798@gmail.com (M.K.); mohsanshah136@gmail.com (S.M.R.S.); m.luqmanuaf@gmail.com (M.L.); iqraifitikhar992@gmail.com (I.I.); 2Department of Agronomy, University of Agriculture, Faisalabad 38000, Pakistan; usmanzulfiqar2664@gmail.com; 3Department of Botany, Government College Women University, Faisalabad 38000, Pakistan; arneebtariq@gcwuf.edu.pk; 4National Institute for Genomics and Advanced Biotechnology (NIGAB), National Agricultural Research Centre (NARC), Park Road, Islamabad 45500, Pakistan; shahzad_nibge77@yahoo.com (S.A.N.); uzairbreeder@gmail.com (M.U.); 5Institute of Crop Sciences, Chinese Academy of Agricultural Sciences, Beijing 100081, China; 6Department of Botany, Shaheed Benazir Bhutto Woman University, Larama, Peshawar 25000, Pakistan; naila.inayat@sbbwu.edu.pk; 7Key Laboratory of Crop Physiology Ecology and Production Management, Ministry of Agriculture, Nanjing 210095, China; attajutt82@yahoo.com

**Keywords:** antioxidants, grain yield, methionine, photosynthesis, wheat

## Abstract

The individual application of pure and active compounds such as methionine may help to address water scarcity issues without compromising the yield of wheat. As organic plant growth stimulants, amino acids are popularly used to promote the productivity of crops. However, the influence of the exogenous application of methionine in wheat remains elusive. The present investigation was planned in order to understand the impact of methionine in wheat under drought stress. Two wheat genotypes were allowed to grow with 100% field capacity (FC) up to the three-leaf stage. Twenty-five-day-old seedlings of two wheat genotypes, Galaxy-13 and Johar-16, were subjected to 40% FC, denoted as water deficit-stress (D), along with 100% FC, called control (C), with and without L-methionine (Met; 4 mM) foliar treatment. Water deficit significantly reduced shoot length, shoot fresh and dry weights, seed yield, photosynthetic, gas exchange attributes except for transpiration rate (*E*), and shoot mineral ions (potassium, calcium, and phosphorus) in both genotypes. A significant increase was recorded in superoxide dismutase (SOD), catalase (CAT), hydrogen peroxide (H_2_O_2_), malondialdehyde (MDA), and sodium ions (Na^+^) due to water deficiency. However, foliar application of Met substantially improved the studied growth, photosynthetic, and gas exchange attributes with water deficit conditions in both genotypes. The activities of SOD, POD, and CAT were further enhanced under stress with Met application. Met improved potassium (K), calcium (Ca^2+^), and phosphorus (P) content. In a nutshell, the foliar application of Met effectively amended water deficit stress tolerance by reducing MDA and H_2_O_2_ content under water deficit conditions in wheat plants. Thus, we are able to deduce a positive association between Met-induced improved growth attributes and drought tolerance.

## 1. Introduction

Around the world, drought stress is considered the detrimental abiotic factor that most severely impairs the growth and development of crops, ultimately causing significant yield reduction. It imparts a significant impact on the cropping area, more than any other kind of abiotic stress [1,2]. Climate severity is expected to be enhanced with climate variation, which may have a significant effect on crop production, primarily due to drought [3]. According to prevailing climatic conditions, it is expected that by the year 2025, 1.8 billion population will face an acute shortage of water [4]. Globally, drought stress has become more intense and is spreading more frequently, and this trend is expected to continue [5].

With the concurrent data on water scarcity issue worldwide, it is approximated that deficiency of irrigation water has already caused a massive reduction in cereal yield (10%) over roughly the previous five decades, and may affect 50% or more of all cultivated land by 2050 [6]. Wheat is the second-most widely cultivated and consumed crop, after only *Zea mays* [7]. Considering ongoing climatic trends, the yield of wheat is continuously affected by environmental constraints, of which drought is the least understood source of damage among the many abiotic stress factors [8]. Previous research has shown that a 56% reduction in wheat production can be attributed to water stress, with the remaining 44% due to other environmental stresses [9]. Wheat is a C_3_ crop, and any fluctuation in the water supply will thus affect wheat growth from moderately to severely [10]. Drought stress causes several negative modifications in the morpho-physiological and metabolic processes of plants [11,12]. Drought induces alterations in water relations, reduces growth, causes ionic and nutrient imbalances, and disrupts the photosynthetic apparatus [13].

Drought stress induces the activation of highly unstable reactive oxygen species, which oxidize the photosynthetic pigments, damage integral proteins and nucleic acids, and disrupt the redox status of cells [14]. Interestingly, plants possess a plethora of mechanisms to mitigate drought-induced oxidative stress by initiating the cellular antioxidant response to scavenge the reactive oxygen species (ROS) and free radicals [15,16]. During drought stress, glyoxalase (a key metalloenzyme in the glycolytic pathway involved in the detoxification of reactive methylglyoxal into d-lactate using glutathione (GSH) as a cofactor) and generation of cellular non-enzymatic and enzymatic antioxidants must be strongly regulated in order to prevent membranes, maintain cell turgor pressure, and protect essential macromolecules and enzymes from oxidation due to overproduction of excited oxygen species [17].

Methionine is an essential acid that takes part in numerous biochemical processes. It is an important as a component of proteins and in the metabolism of carbon, and its sulfur-bound methyl group works as a precursor to activate S-adenosylmethionine for methane production [18]. Methionine ameliorates drought stress by improving antioxidants (CAT, POX, SOD) and by increasing proline and photosynthetic pigments concentration [19].

To the best of our knowledge, this study is the first detailed study of the impact of methionine on wheat growth and yield, the antioxidant defense system, osmolytes, ROS, and shoot ionic content. The present study can open new horizons for understanding the physiological impact of methionine in response to water deficit conditions.

## 2. Materials and Methods

### 2.1. Plant Source, Experimental Layout, and Establishment

Two wheat genotypes, Galaxy-13 and Johar-16, were procured from Ayub Agricultural Research Institute (AARI), Faisalabad, Pakistan. Healthy seeds were selected based on size homogeneity and soaked in 30% (*v*/*v*) H_2_O_2_ for 5 min to sterilize them and subsequently washed thrice with deionized water. After sterilization, the seeds were immersed in distilled water (H_2_O) for 24 h and dried at room temperature. After complete drying, the sterilized seeds were laid down in round plastic buckets (depth 50 cm; circumference 30 cm). The buckets were topped up with 4 kg of soil and sand. Pots were set up as a CRD (Completely Randomized Design) in distinct rows and columns with three replicates. The day and night temperatures were maintained at 24 ± 1 and 18 ± 1 °C (13/11 h), respectively, which is optimal for plant growth in average growing environments found in Pakistan. Average relative humidity was maintained at 50–55% and 75–80% for day and night, respectively.

#### 2.1.1. Water Deficit Stress Imposition and Methionine Applications

After 25 days of germination, seedlings were dissected as control (C) by keeping field capacity (FC) 100% and 35–40% as water deficit stress (D). Field capacity (FC) was maintained on a daily basis by weighing the plastic pots and maintaining the lost water by pouring in an equivalent quantity of water. Field capacity was regularly monitored before each water application using the following equation:$$W=D\times H\times A\;\left({\mathrm{FC}}_{1}-{\mathrm{FC}}_{0}\right)$$
where W = the amount of water provided to the pot, D = the density of soil in the pot, H = soil depth, A = area (L × W) of the pot, FC_1_= FC desired, and FC_0_ = FC before watering.

Control (C) and water deficit (D)-treated plants were again dissected for foliar application of 4 mM methionine (Met). L-methionine was weighed and dissolved in distilled water. The desired millimolar (mM) solution was mixed with 1 mL tween-20 as surfactant. The complete layout of the experiment was: (i) Control (C); (ii) water deficit (D); (iii) water deficit + Methionine (D + Met); and (iv) Methionine (Met.). All treatments were repeated thrice.

#### 2.1.2. Plant Sampling

The plants were harvested from the experimental pots at the physiological maturity stage (growth stage 8 with BBCH code 89), roughly ~155 days after sowing (DAS). Data regarding growth traits were recorded following standard procedures. Shoot/root fresh weights (SFW) were weighed with an analytical balance immediately after harvesting. Later, shoot/root samples were transferred to a 70 °C oven to record the dry weight of the samples. Shoot/root length was measured with a measuring tape (cm).

### 2.2. Plant Analysis and Measurements

#### 2.2.1. Gas Exchange Traits

The physical flow of gases of different treatments were monitored in terms of the CO_2_ fixation rate (*A*), evaporation rate (*E*), stomatal regulation/conduction (*g*_s_), and interior CO_2_ concentration (*C*i) of fully matured leaf samples with IRGA (LCA-4, ADC, Hoddesdon, England). All data were intentionally taken when plants were exposed to moderately intense sunlight conditions (9 a.m.–11:45 a.m.). The day was full sunny, with a CO_2_ concentration of 400 μmol mol^−1^ [20,21]. The water use efficiency (WUE) was calculated by applying the formula below.
$$\mathrm{WUE}=\frac{{\mathrm{CO}}_{2}\;\mathrm{assimilated}}{\mathrm{transpiration}}$$

#### 2.2.2. Photosynthetic Pigments

For Chlorophyll *a*, *b*, total chlorophyll content, and carotenoids, leaf samples were homogenized in 80% acetone [22]. Homogenized plant crude extract was centrifuged for 10 min at 12,000 rpm. The optical density of transparent filtrate was determined at 663 nm and 645 nm with a UV-visible spectrophotometer (IRMECO 2020) [23].

### 2.3. Determination of Cellular Antioxidants

The fresh leaf sample was finely ground in 50 mM ice-cooled potassium phosphate buffer (pH 7.8). The crude homogenate was centrifuged at 12,000 rpm for 20 min. The enzyme extract was preserved at 4 °C for 36 h for the determination of antioxidants, soluble proteins, and total free amino acids.

#### 2.3.1. Soluble Proteins

The content of soluble proteins was assayed using the Bradford method [24]. The enzyme extract (100 µL) was mixed with freshly prepared Bradford regent (5 mL), then kept at room temperature for 5 min. The optical density was recorded at 595 nm with a UV-Visible spectrophotometer (IRMECO 2020).

#### 2.3.2. Peroxidase (POX) Activity

POX was found spectrophotometrically by the Goliber method [25]. Guaiacol was oxidized in the presence of H_2_O_2_ and expressed as mg^−1^ min^−1^. The reaction contained 50 µL of the enzyme extract, 20 mM guaiacol, 10 mM H_2_O_2_, and 50 mM potassium phosphate buffer (pH 7.8). The enzyme efficiency was calculated by measuring at 460 nm for 2 min. with an interval of 30 s using a spectrophotometer (IRMECO 2020). The enzyme-specific activity was articulated on the basis of proteins, shown as Units mg^−1^ Protein.

#### 2.3.3. Catalase (CAT) Activity

Catalase activity was determined according to the method in [26]. The breakdown of H_2_O_2_ was monitored during the enzyme assay. Enzyme extract 200 µL was added to 2 mL of reaction mixture containing potassium phosphate buffer (50 mM) and H_2_O_2_ (30 mM). The breakdown of H_2_O_2_ was measured by the decrease in optical density at 240 nm with a UV-visible spectrophotometer (IRMECO 2020).

#### 2.3.4. Superoxide Dismutase (SOD) Activity

Superoxide dismutase was determined by following the novel methodology of [27]. The SOD activity was monitored in 3 mL total volume by adding NBT, riboflavin, methionine, EDTA, phosphate buffer (pH 7.8), and enzyme extract. Test tubes containing the reaction solution were illuminated under white fluorescent light for 20 min. The absorbance of the solution was recorded at 560 nm with a spectrophotometer. The activity of the enzyme was expressed as Units mg^−1^ Protein.

### 2.4. Evaluation of Stress Markers (MDA and H_2_O_2_)

The extent of oxidative stress was assayed from the wheat seedlings in terms of malondialdehyde (MDA) and hydrogen peroxide (H_2_O_2_) by the [28,29]. In order to determine MDA contents, 0.25 g of the fresh leaf sample of each treatment was ground in 5 mL trichloroacetic acid (TCA; 1%) and centrifuged at 15,000 rpm for 15 min. The supernatant was transferred to a glass test tube and mixed with TCA and thiobarbituric acid (TBA) solution. The glass tubes with their constituents were placed in a water bath at 95 °C for 25–30 min. The samples were removed from the water bath and the reaction was immediately terminated in ice. The optical density (OD) was monitored at 532 nm and 600 nm using a UV-visible spectrophotometer (IRMECO 2020). The specific value of absorption at 532 nm was subtracted from the non-specific absorption value of 600 nm and expressed as nmol g^−1^ FW. For determination of H_2_O_2_ concentration in all stressed and non-stressed treatments and replicates, 500 mg of plant sample was ground in 5 mL of 0.1% TCA. Later, 0.5 mL aliquots were taken and mixed with 0.5 mL of 50 mM potassium phosphate buffer (pH 7.6) and 1 mL of 1 M potassium iodide (KI). The absorbance was taken at 390 nm with the help of an IRMECO-2020 spectrophotometer, expressed as µmol^−1^ g^−1^.

### 2.5. Shoot Ionic Contents

The shoot samples were dried at 70 °C in a drying oven for 72 h, and dried shoot biomass (500 mg) was digested with H_2_SO_4_ in a 50 mL digesting flask. After overnight incubation, digesting flasks were kept on the heating block (350 °C) until fumes started evaporating, then 35% H_2_O_2_ was slowly added to obtain a colourless transparent solution. Later, the solution was diluted to 50 mL with distilled water and filtered. The filtrate samples were subjected to ionic (Na^+^, Ca^2+^, P, and K^+^) analysis with a photometer (Jenway, PFP-7) [30].

### 2.6. Yield Assessment

Wheat plants were harvested after 75 days of sowing at the grain stage. The yield attributes, such as number of grains per plant (NGPP), grain yield per plant (GYPP), and hundred-grain weight (HGW), were calculated with standard protocols.

### 2.7. Statistical Analysis

Data from three randomly-selected replicates were considered for analysis of variance (ANOVA) with Minitab 19 at *p* < 0.05 significance level. Additionally, the pairwise difference was calculated using Tukey’s test. Scientific findings were subjected to principal component analysis (PCA) and cluster heatmaps by advance data analysis software (R Core Team, 2019: R Foundation for Statistical Computing, Vienna, Austria) in order to monitor the correlations between different studied features.

## 3. Results

### 3.1. Enhancement of Biomass of Wheat with Met under Water Deficit Stress

Methionine application under non-stress conditions significantly influenced (*p* ≤ 0.05) biomass traits, i.e., shoot length (SL), shoot fresh weight (SFW), shoot dry weight (SDW), root length (RL), root fresh weight (RFW), and root dry weight (RDW), of two wheat genotypes (Table 1). Both genotypes exhibited an increase in SL, SFW, SDW, RL, RFW, and RDW with Met compared to water deficit stress (D). It was observed that foliar application of Met increased SL up to 5.54%, SFW 30.98%, SDW 20%, RL 32.78%, RFW 34.86%, and RDW 20.29% with 40% FC in Galaxy-13. However, under stress conditions (D), both genotypes showed a significant reduction in biomass. Moreover, Met application under non stress conditions (FC 100%) improved the growth attributes compared to control plants (Table 1). A clear difference exists between the two wheat cultivars, with Johar-16 being more sensitive to water deficit stress.

### 3.2. Enhancement of Yield Traits of Wheat with Met under Water Deficit Stress

Under water deficit stress, yield attributes, i.e., the number of grains/plant (NGPP), grain yield/plant (GYPP), and hundred-grain weight (HGW) were reduced significantly (*p* ≤ 0.05) (Table 1). Methionine application remarkedly improved the yield traits under water stress. There was a ~6–11% increase of all yield traits with Met application under water deficit stress, with Galaxy-13 showing a higher yield compared to Johar-16 (Table 1).

### 3.3. Enhancement of Gas Exchange Traits of Wheat with Met under Water Deficit Stress

Imposition of water deficit stress slowed net CO_2_ assimilation rate (*A*), stomatal conductance (*g*_s_), sub-stomatal CO_2_ concentration (*C*i), and water use efficiency (A/E) in both genotypes compared to control, irrespective of methionine application (Table 1). Methionine application under stress (7.16%) and non-stress (28.26%) conditions considerably increased the net CO_2_ assimilation rate (35.15%) in both genotypes. A marked increase in transpiration rate (*E*) was observed with 40% FC. However, Met application significantly reduced *E* in both stress (24%) and non-stress conditions (43%) (Table 1). The *g*_s_ (35.82%), *C*i (5.22%), and WUE (35.23%) decreased with limited water supply in both wheat genotypes. However, all gas exchange attributes improved with foliar application of Met under stress (Table 1).

### 3.4. Enhancement of Photosynthetic Pigments of Wheat with Met under Water Deficit Stress

Application of Met under both water deficit stress and non-stress conditions influenced chlorophyll and carotenoid contents significantly (*p* ≤ 0.05) for (Table 1). Imposition of water deficit stress caused a significant decline (*p* ≤ 0.05) in the Chl *a* (45%), *b* (28%)*,* and carotenoid (24%) contents. Application of Met in wheat genotypes under both water deficit and normal conditions showed a significant increase in Chl and carotenoids; however, the increase was more pronounced in plants treated with Met under normal conditions (Table 1).

### 3.5. Enhancement of Soluble Proteins and Cellular Antioxidant Enzymes of Wheat with Met under Water Deficit Stress

Water deficit stress resulted in a significant (*p* ≤ 0.05) increase in soluble proteins in both wheat genotypes; however, Met. supplementation significantly enhanced the accumulation of proteins in both stressed and non-stressed plants (Table 2). The plants treated with the Met under control conditions showed more accumulation of soluble proteins (Table 2).

Water deficit stress markedly decreased the activity of peroxidase (POX) compared to control and counterpart treatments (Table 2). Plants treated with Met under non-stress conditions showed more activity of POX (39%) in comparison to plants under water deficit conditions without foliar application. Applications of Met effectively improved the concentration of enzymatic antioxidants under water deficit and normal irrigation conditions in both wheat genotypes. A highly significant (*p* ≤ 0.01) varietal difference was observed in both wheat genotypes, with Galaxy-13 responding much better to Met treatment than Johar-16 (Table 2). The other antioxidant enzymes, CAT and SOD, were significantly influenced by water deficit and foliar application of Met as well (Table 2). CAT activity was enhanced by up to 6.75% and 12.33% with foliar application of Met under water deficit stress compared to non-treated stressed Galaxy-13 and Johar-16 plants, respectively, while SOD activity decreased by up to 3.32% and 2.09%, respectively, under stress. Again, Met application mitigated the effects of stress and improved the SOD concentration (Table 2).

### 3.6. Reduction of Oxidative Stress Markers (H_2_O_2_ and MDA) in Wheat Treated with Met under Water Deficit Stress

Under water deficit stress, production of H_2_O_2_ was at its maximum, while Met treatments significantly reduced the production of H_2_O_2_ in both wheat genotypes. Galaxy-13 and Johar-16 showed ~20–32% increase in MDA concentration under water deficit stress. The maximum reduction in H_2_O_2_ content was observed in non-stressed plants with Met supplementation (Table 2). MDA activity with 40% FC was substantially (*p* ≤ 0.05) increased in both genotypes due to lipid peroxidation. However, the application of Met under both water deficit and normal condition significantly reduced oxidative damage by reducing the concentration of the MDA content (51–63%) in both wheat genotypes. In this regard, Johar-16 was more drought-sensitive than Galaxy-13 (Table 2).

### 3.7. Enhancement of Shoot Ionic Contents of Wheat with Met under Water Deficit Stress

Under water deficit stress, there was more accumulation of Na^+^ in both genotypes, while Met application significantly reduced Na^+^ under both stress (21.73%) and normal conditions (20.37%). The concentration of K^+^ and Ca^2+^ was significantly increased by Met application under water deficit as well as in non-stressed plants. Neither water deficit stress nor Met supplementation had a significant impact on the P content of either studied genotypes (Table 2).

### 3.8. Principal Component Analysis (PCAs)

Ellipsed PCAs were constructed (*p* ≤ 0.05) for growth, gas exchange, and photosynthetic pigments in order to evaluate the principal components for water deficit and methionine treatments. Two principal components (PC1, 75.3% and PC2, 23.6%) with a total variance of 98.9% were selected from among 24 indices. Both of the genotypes G1 and G2 ellipsed together and showed a strong influence on each other at varying levels of the treatments. The transpiration rate (*E*) was significantly increased under water deficit (D), with higher positive eigenvalues (+3). The growth traits, such as SFW, SDW, and SL, showed a significantly higher increase under absolute control (C) and medium influence of water deficit and methionine (D + Met) supplementation. Photosynthetic traits, such as *A*, *g*_s_, *A/E* and Chl *a,* Chl *b*, T. Chl, and Caro, showed significantly lower negative eigenvalues under methionine individual treatment in both genotypes (Figure 1A).

PCAs were constructed between yield and biochemical traits of two wheat genotypes and significantly (*p* ≤ 0.05) corresponding variations as PC1 62.4% and PC2 27.9% (Total 90.3%). The combined exogenous applications of water deficit and methionine (D + Met.) showed a significant positive influence on antioxidants, such as CAT, SOD, and POX, with higher positive eigenvalues (>4). The yield traits HSW, NSPP, and SYPP significantly corresponded under individual applications of methionine. The combined applications of methionine and water deficit reduced MDA content in both genotypes. The concentration of H_2_O_2_ was increased under water deficit, with negative eigenvalues (Figure 1B).

### 3.9. Clustered Heatmap Analysis

A clustered heatmap was constructed among growth, gas exchange, and photosynthetic pigments in order to demonstrate the contribution of water deficit tolerance among both wheat genotypes. The SFW and SDW were closely associated with the G1 and methionine applications, and were thus strongly clustered. Photosynthetic traits such as *A*, *A/E*, *g*_s_, and Chl *a*, *b* were strongly and significantly increased under control (C) and methionine (Met) treatment, which influenced the water deficit tolerance. All of these traits were negatively associated under the influence of water deficiency, showing significant reductions. Strong clustering was found between both genotypes in response to water deficit as an individual treatment (Figure 2A). The heatmap provides a global perspective on water deficit tolerance between both genotypes regarding yield and biochemical traits. The organic osmolyte TSP was significantly correlated with all yield traits, and showed an increasing pattern in both genotypes in response to Met treatment. The antioxidants (SOD, CAT, and POX) were significantly associated with Met applications under water deficit stress (D + Met). The increase in cellular enzyme activities greatly contributed to the drought tolerance efficiency of both genotypes. Water deficit stress treatment (D) caused a significant decline in the concentration of cellular enzymes (Figure 2B).

## 4. Discussion

Water is the most indispensable molecule for plants’ metabolic processing, including photosynthesis, enzymatic functioning, water, nutrient balance, etc. [31,32]. *Triticum aestivum* L., a premeditated cereal C_3_ crop grown around the globe, is adapted to grow under a variety of climatic variations and plays a key part in maintaining human nutrition requirements [33]. The findings of the current experiment advocate for the significance of methionine in boosting the growth and other vital cellular processes of wheat (*Triticum aestivum* L.) seedlings suffering from water-deficient condition. In the current study, both genotypes showed an increase in SL, SFW, SDW, RL, RFW, and RDW with Met application as compared to water deficit stress (D) without treatment. Our results show that foliar application of Met increased SL up to 5.54%, SFW 30.98%, SDW 20%, RL 32.78%, RFW 34.86%**,** and RDW 20.29%, with 40% FC in Galaxy-13. The use of exogenously applied plant growth regulators is a significant technique for overcoming the adverse effects of water limitation [34]. In the last decade, a large number of *MSR* (Methionine sulfoxide reductase) genes have been isolated, cloned, and characterized in many species, from algae to wheat [31,35]. Particularly, *TaMSRB3.1* (*Triticum aestivum Methionine Sulfoxide Reductase*) is expressed in the chloroplasts and has shown a tendency to reduce MSR to Met in wheat. Under osmotic stress, *TaMSRB3.1* overexpression in the wheat plant has been described in terms of its defensive roles in ROS overaccumulation, where it activates the abscisic acid (ABA) signaling pathway in order to limit the rate of transpiration [36]. Similarly, in rice, the application of sucrose increases tiller development [37]. A limited supply of water is a very significant restraining factor, particularly in the early growth stage of plant establishment, resulting in abnormalities of many physio-biochemical traits of wheat plants [12,38]. Similar results were observed in this study; see Table 1 and Table 2 and Figure 1 and Figure 2. Water deficit stress significantly reduced all growth traits, with negative consequences on the yield and productivity of both of the studied wheat cultivars. However, the exogenous application of Met significantly enhanced all studied features of growth under water stress (Table 1) compared to control; the same result has been reported in maize under salt stress [39,40]. Met has been found to alter metabolic processes in plants experiencing saline conditions [41]. The exogenous application of proline and Met additionally enhanced growth, productivity, ad non-enzymatic and enzymatic antioxidants in cowpea plants subjected to water stress [19]. Moreover, the application of methionine, lysine, and tyrosine was considered advantageous for shoot growth, *A*, and water use efficiency (WUE). It was concluded that treatment could result in plants having superior reduction power, vitality, and carbon molecules to harness growth activities [42].

It has been reported that during drought stress, crop plants such as cereals reduce their photosynthetic rate [43]. In the current study, photosynthetic and gas exchange attributes were substantially decreased during the water deficit period, and these changes were reversed with methionine application (Table 1). Previously, it has been reported in many crops, including wheat, that chlorophyll content was reduced under drought stress [21,44]. Here, Met treatment significantly contributed to higher *Chl a*, *b,* and carotenoid content (Table 1). Carotenoids, as non-enzymatic antioxidants, minimized the photooxidative destruction caused by regulation stress-related proteins and osmolytes, thereby enhancing growth processes [45]. Previously, in wheat, rice, and peanut plants it has been reported that *g*_s_ and *E* have a strong correlation [46,47]. It is obvious from Table 1 that photosynthetic modulation due to water deficit stress in both wheat cultivars governs the mechanism of stomatal conductance (*g*_s_), which in turn contributes toward decreased CO_2_ availability for CO_2_ assimilation (*A*). Similarly, the application of methionine may be involved in the regulation of water status to maintain *g*_s_ and increase in stomatal aperture to facilitate diffusion of CO_2_ from the atmosphere to the carboxylation site of Rubisco [48]. It has been observed that the CO_2_ pool drastically decreases due to overaccumulation of ROS [32]. We monitored a disproportionate ration of H_2_O_2_ to its antioxidant pool (Table 1) which was positively correlated to MDA concentration, which is a consistent stress markers in all plant species under unfavorable environmental conditions [49]. The methionine biosynthetic pathway has a certain relation to photorespiration via serine and 1-carbon metabolism, as described for wheat in [50], and its biosynthesis under high photorespiration without water stress has been studied in *Helianthus annuus* and *Arabidopsis thaliana* [51]. Our findings suggest that with respect to overaccumulation of ROS and lipid peroxidation with water stress. (Table 2), the application of Met regulates ROS through enhanced activities of antioxidant enzymes (Table 2) in stressed plants. Met serves as a precursor of Cys, which is involved as an intermediate in glutathione synthesis and has outstanding antioxidant activity [52]. Foliar application of Met evolves tolerance mechanisms in tomato plants via concerted regulation of its antioxidant pool, osmoprotectants, reduced lipid peroxidation, and shielding of photosynthetic pigments, ultimately refining growth characteristics [48]. The observations in the current experiment demonstrate that water scarcity substantially reduced nutrient uptake, whereas Met reduced the negative impact of water stress by improving the uptake of essential ions (K^+^, Ca^2+^) in both cultivars of wheat (Table 2). The reduction in chlorophyll content due to water stress may be due to the disorganization of the thylakoid membrane, along with damaging effects on chlorophyllase enzymes, which in turn inhibit ion accumulation [53]. Met fortification with phosphate has previously been found to cause a many-fold increase in the yield of the essential oil of valerian, supporting the current findings that the increase in yield thanks to Met can considerably increase the yield and quality of wheat [54]. The current findings prove the interaction of water deficit and Met in improving uptake of N, P, and K in wheat seedlings, as previously found in cowpea plants [19].

## 5. Conclusions

Exogenous methionine application partially or completely alleviated the drought-induced inhibition of whole-plant growth and antioxidant metabolism by reducing lipid peroxidation and ROS. Methionine is speculated to be involved in mitigating water stress in wheat seedlings, as depicted in the current findings. Exogenous methionine subsequently enhances the biosynthesis of the endogenous antioxidant pool. Methionine suppressed the MDA and ROS activity in wheat seedlings. Methionine represents a promising candidate agent for use in crop improvement and protection against water scarcity conditions.

## Figures and Tables

**Figure 1 life-12-00969-f001:**
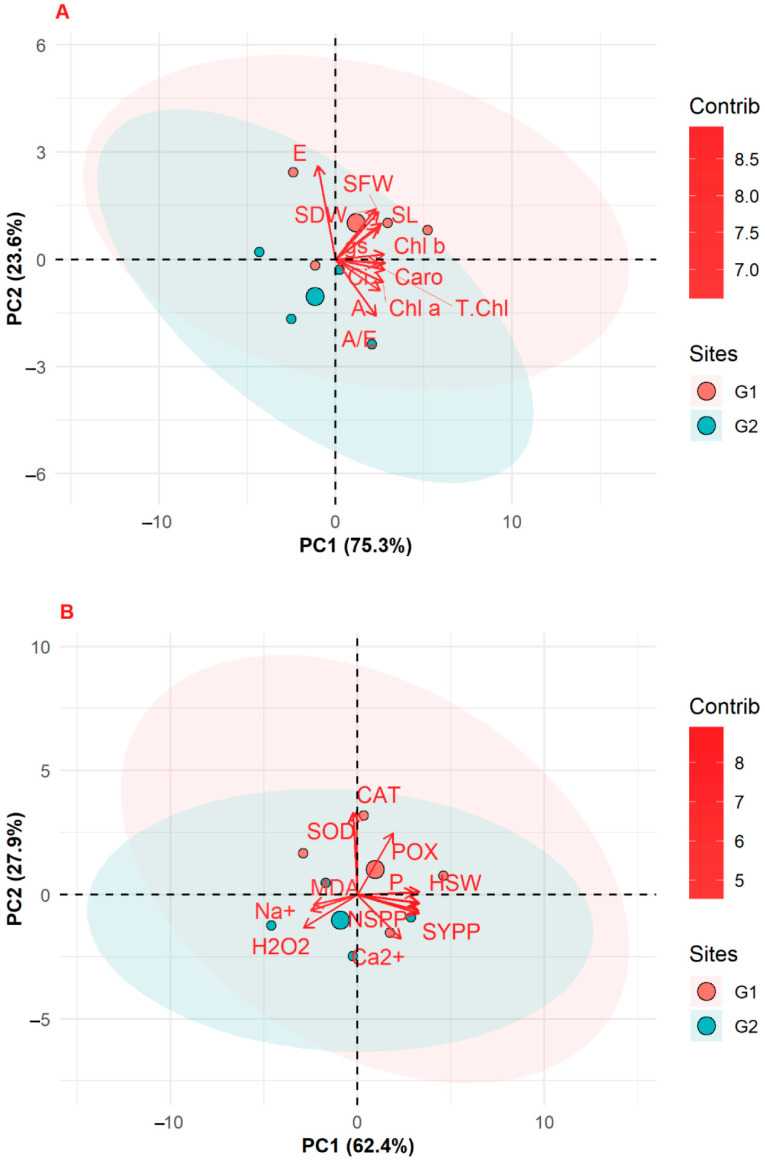
(**A**) PCA biplot for growth, chlorophyll content, and gas exchange and (**B**) yield, osmolytes, ROS, and antioxidant enzymes for the two wheat genotypes under water deficit stress and methionine treatment.

**Figure 2 life-12-00969-f002:**
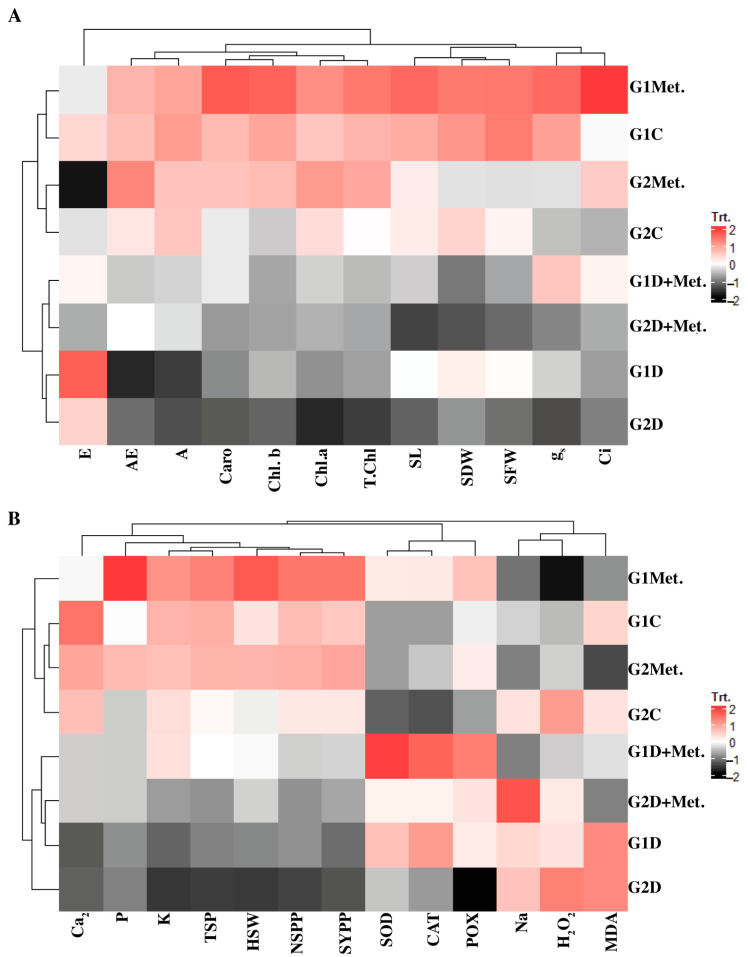
(**A**) Clustered heatmap among growth, chlorophyll content, and gas exchange and (**B**) yield, osmolytes, ROS, and antioxidant enzymes for the two wheat genotypes under water deficit stress and methionine treatment.

**Table 1 life-12-00969-t001:** Growth, yield, gas exchange, and photosynthetic pigments as influenced by methionine under water deficit stress in two wheat genotypes.

		Galaxy-13				Johar-16			
	Traits	Control	Water Deficit	Water Deficit + Met.	Methionine (Met)	Control	Water Deficit	Water Deficit + Met.	Methionine (Met)
**Growth traits**	SL	62.00 ± 2.40 b	51.50 ± 1.71 c	55.58 ± 1.57 c	70.05 ± 1.75 a	54.33 ± 1.23 a	41.35 ± 1.50 b	48.73 ± 1.32 b	54.05 ± 1.52 a
	SFW	9.18 ± 0.22 a	5.68 ± 0.24 b	8.23 ± 0.31 c	9.58 ± 0.44 a	8.80 ± 0.25 a	3.33 ± 0.20 b	6.25 ± 0.18 b	6.98 ± 0.22 a
	SDW	3.30 ± 0.16 a	1.88 ± 0.17 ab	2.35 ± 0.16 b	3.43 ± 0.23 a	2.00 ± 0.18 a	1.45 ± 0.16 ab	1.23 ± 0.12 b	1.70 ± 0.17 ab
	RL	18.31 ± 1.17 a	10.17 ± 2.01 c	15.13 ± 1.14 b	20.17 ± 1.87 a	17.19 ± 1.28 b	11.39 ± 1.02 d	14.14 ± 1.09 c	19.41 ± 2.17 a
	RFW	4.61 ± 0.19 a	2.13 ± 0.09 c	3.27 ± 1.09 b	4.78 ± 1.19 a	4.01 ± 0.41 b	2.01 ± 0.37 c	2.35 ± 1.21 c	4.89 ± 1.09 a
	RDW	1.77 ± 0.04 a	1.10 ± 0.00 c	1.38 ± 0.01 b	1.73 ± 0.07 ab	0.98 ± 0.00 b	0.04 ± 0.01 d	0.59 ± 0.05 c	1.10 ± 0.09 a
**Yield traits**	NGPP	197.00 ± 1.58 ab	177.00 ± 1.47 c	190.00 ± 1.47 b	202.00 ± 1.29 a	194.00 ± 0.91 a	170.00 ± 1.47 c	187.00 ± 0.91 b	198.00 ± 1.47 a
	GYPP	8.91 ± 0.06 ab	7.96 ± 0.02 c	8.76 ± 0.02 b	9.09 ± 0.03 a	8.81 ± 0.02 a	7.56 ± 0.02 c	8.31 ± 0.02 b	8.96 ± 0.02 a
	HGW	4.60 ± 0.09 a	3.75 ± 0.15 c	4.26 ± 0.17 b	4.63 ± 0.09 a	4.03 ± 0.23 b	3.60 ± 0.12 c	3.80 ± 0.11 b	4.21 ± 0.08 a
**Gas exchange traits**	*A*	9.40 ± 0.48 a	5.38 ± 0.43 c	8.30 ± 0.29 b	9.35 ± 0.40 a	7.05 ± 0.24 b	5.50 ± 0.46 c	8.38 ± 0.24 ab	9.08 ± 0.23 a
	*E*	0.45 ± 0.09 ab	0.55 ± 0.02 a	0.41 ± 0.03 b	0.31 ± 0.06 c	0.41 ± 0.02 b	0.50 ± 0.02 a	0.36 ± 0.02 c	0.38 ± 0.02 c
	*g* _s_	85.75 ± 1.89 b	52.75 ± 1.49 c	94.88 ± 1.78 a	97.00 ± 1.96 a	82.50 ± 1.71 b	70.50 ± 2.40 c	91.50 ± 1.44 a	93.00 ± 1.08 a
	*C*i	225.75 ± 3.68 b	211.00 ± 3.16 c	269.75 ± 1.75 b	275.00 ± 2.08 a	214.50 ± 4.41 b	206.00 ± 2.27 c	243.50 ± 2.90 a	239.50 ± 2.90 a
	WUE	20.40 ± 3.02 b	13.38 ± 0.32 c	27.11 ± 0.53 a	20.72 ± 3.22 a	19.12 ± 0.73 b	15.01 ± 1.15 c	18.28 ± 0.92 b	22.17 ± 1.10 a
**Photosynthetic pigments**	Chl *a*	1.10 ± 0.09 b	0.60 ± 0.18 c	1.40 ± 0.16 a	1.53 ± 0.13 a	1.35 ± 0.10 b	0.93 ± 0.13 c	1.15 ± 0.13 ab	1.50 ± 0.17 a
	Chl *b*	0.95 ± 0.06 b	0.68 ± 0.09 c	1.65 ± 0.06 a	1.08 ± 0.09 a	0.70 ± 0.07 b	0.62 ± 0.03 c	0.75 ± 0.02 ab	0.90 ± 0.13 a
	T. Chl	1.75 ± 0.09 ab	0.78 ± 0.15 c	1.85 ± 0.16 ab	2.60 ± 0.17 a	2.05 ± 0.15 b	1.48 ± 0.16 c	2.43 ± 0.13 a	2.40 ± 0.26 a
	Caro.	0.81 ± 0.08 ab	0.61 ± 0.07 b	0.70 ± 0.08 ab	0.94 ± 0.07 a	0.70 ± 0.07 b	0.56 ± 0.02 c	0.83 ± 0.02 b	0.97 ± 0.04 a

Values are expressed as means ± SE of four replicates. Different lowercase letters following the data within the same rows indicate a significant difference at *p* ≤ 0.05. SL—shoot length (cm), SFW—shoot fresh weight (g plant^−1^), SDW—shoot dry weight (g plant^−1^), RL—root length (cm), RFW—root fresh weight (g plant^−1^), RDW—root dry weight (g plant^−1^), NGPP—number of grains per plant^−1^, GYPP—grain yield per plant^−1^, HGW—hundred grain weight (g), *A*—net CO_2_ assimilation rate (µmol CO_2_ m^−2^ s^−1^), *E*—transpiration rate (mmol H_2_O m^−2^ s^−1^), *g*_s_—stomatal conductance (mmol m^−2^ s^−1^), *C*i—sub-stomatal CO_2_ concentration, WUE—water use efficiency, Chl *a*—chlorophyll *a* (mg g^−1^ FW), Chl *b*—chlorophyll *b* (mg g^−1^ FW), T. Chl—chlorophyll *a + b* (mg g^−1^ FW), Caro—carotenoids (mg g^−1^ FW).

**Table 2 life-12-00969-t002:** Compatible solutes, antioxidant enzymes, reactive oxygen species (ROS), malondialdehyde (MDA), and shoot ionic content as influenced by methionine under water deficit stress in two wheat genotypes.

	Galaxy-13				Johar-16			
Traits	Control	Water Deficit	Water Deficit + Met.	Methionine (Met)	Control	Water Deficit	Water Deficit + Met.	Methionine (Met)
**Compatible solute**								
TSP	2.88 ± 0.13 c	2.80 ± 0.15 b	3.59 ± 0.14 a	3.65 ± 0.26 a	2.18 ± 0.27 c	2.60 ± 0.18 bc	2.85 ± 0.21 ba	3.45 ± 0.25 a
**Antioxidants enzymes**								
POX	12.30 ± 0.53 b	8.75 ± 0.47 c	12.35 ± 0.38 b	15.30 ± 0.38 a	9.55 ± 0.28 b	9.90 ± 0.38 c	12.85 ± 0.41 ab	14.73 ± 0.26 a
SOD	4.10 ± 0.20 b	4.83 ± 0.22 ab	5.50 ± 0.17 a	6.60 ± 0.30 a	3.95 ± 0.17 b	4.05 ± 0.17 b	4.25 ± 0.16 ab	4.90 ± 0.20 a
CAT	9.15 ± 0.53 b	11.73 ± 0.48 ab	12.58 ± 0.44 a	13.53 ± 0.36 a	8.28 ± 0.48 b	9.10 ± 0.53 b	10.38 ± 0.26 a	10.58 ± 0.31 a
**ROS**								
H_2_O_2_	50.53 ± 0.93 b	63.60 ± 1.46 a	51.03 ± 0.90 b	45.43 ± 1.11 c	50.51 ± 1.16 b	71.65 ± 0.84 a	48.33 ± 0.83 bc	31.10 ± 0.92 c
MDA	16.09 ± 0.65 b	22.58 ± 0.40 a	13.60 ± 0.64 bc	8.45 ± 0.34 c	20.73 ± 0.62 ab	26.58 ± 0.41 a	15.23 ± 0.53 c	14.35 ± 0.39 c
**Shoot ionic contents**								
Na^+^	4.18 ± 0.27 a	4.63 ± 0.38 a	3.81 ± 0.12 b	3.75 ± 0.2 b	4.56 ± 0.21 a	4.75 ± 0.32 a	5.44 ± 0.21 b	3.81 ± 0.28 c
K^+^	5.65 ± 0.30 b	5.15 ± 0.18 c	6.30 ± 0.18 ab	6.93 ± 0.22 a	6.31 ± 0.33 a	4.88 ± 0.15 c	6.38 ± 0.23 a	6.55 ± 0.21 a
Ca^2+^	11.55 ± 0.21 a	6.25 ± 0.32 c	10.00 ± 0.46 b	10.25 ± 0.32 ab	10.88 ± 0.43 ab	9.30 ± 0.29 b	10.00 ± 0.46 ab	11.13 ± 0.43 a
P	0.10 ± 0.01 a	0.08 ± 0.02 a	0.09 ± 0.01 a	0.14 ± 0.01 a	0.09 ± 0.01 a	0.08 ± 0.01 a	0.09 ± 0.01 a	0.11 ± 0.01 a

Mean values and ±SE of four replicates are provided. Different lowercase letters following the data within the same rows indicate significant difference at *p* ≤ 0.05. TSP—total soluble proteins (mg g^−1^ FW.), POX—peroxidase (units mg^−1^ protein), SOD—superoxide dismutase (units mg^−1^ protein), CAT—catalase (units mg^−1^ protein), H_2_O_2_—Hydrogen peroxide (µmol g^−1^ FW), MDA—malondialdehyde (nm g^−1^ FW), Na^+^—sodium ions (mg g^−1^ DW), K^+^ (mg g^−1^ DW), Ca^2+^—calcium ions (mg g^−1^ DW), P—phosphorus ions (mg g^−1^ DW).

## Data Availability

All data related to this study are presented in the main text.

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
