# Peer review of "Methionine Promotes the Growth and Yield of Wheat under Water Deficit Conditions by Regulating the Antioxidant Enzymes, Reactive Oxygen Species, and Ions"

_life, 2022, doi:10.3390/life12070969_

Round 1

Reviewer 1 Report

In the present study entitled  Methionine Promotes the Growth and Yield of Wheat under Water Deficit Conditions by Regulating the Antioxidants Enzymes, Reactive Oxygen Species, and Ions. Maqsood et al. study provided evidence that Methioninec can promote the yield parameters under water deficit stress. Overall, the manuscript is well written and the data presented is convincible. I have some suggestions/comments on it as follows:

·       Line 34 and Line 38, SOD and POD have been given to their full form at both places in the abstract, give them only once. Authors should also check the whole manuscript for such minor grammar mistakes.

·       Authors should differentiate, whether they studied and reported the phenotype under water deficit stress or drought stress. Technically water deficit stress and drought stress are relevant but not the exactly same.

·       Authors used two different formulas for hydrogen peroxide (H2O2) and H2O2, I will suggest to the authors please confirm it and used the correct one.

·       The authors used L-methionine (Met; 4mM) as a foliar treatment. On which bases they selected this dose, please provide references from the previous study if any

·       Similarly, two wheat verities “Galaxy-13 and Johar-16” were used in this study. Why only these two verities were used.

·       Line 85: “Heathy seeds were selected based” there is a spelling mistake, please confirm.

·       Sterilized seeds were dried, but authors didn’t mention how they were dried.

·       In some places, authors added space and some places not, e.g., (deep; 50 cm, and circumference; 30cm). I will suggest to them please use one format.

·       Authors measured the water use efficiency (WUE) by applying a formula. Please provide detail formula.

·       Line 189: Please confirm the sentence, “hundredd grain weight (HGW) were calculated with standard protocols.

·       Line 199: P values should be in italic form.

·       Line 214: please confirm the units of chlorophyll.

·       Please improve the discussion part by comparing the previous research in other crop species.

Author Response

In the present study entitled  “Methionine Promotes the Growth and Yield of Wheat under Water Deficit Conditions by Regulating the Antioxidants Enzymes, Reactive Oxygen Species, and Ions”. Maqsood et al. study provided evidence that Methioninec can promote the yield parameters under water deficit stress. Overall, the manuscript is well written and the data presented is convincible. I have some suggestions/comments on it as follows:

Dear Reviewer, Thank you for your careful reading of our manuscript and your helpful comments and questions. We have carefully revised this manuscript according to yours comments. A point-by-point reply to these comments is below:

  • Line 34 and Line 38, SOD and POD have been given to their full form at both places in the abstract, give them only once. Authors should also check the whole manuscript for such minor grammar mistakes.

Response: Thank you for such a nice suggestion. We have provided full form at first appearance and rest of the places we used abbreviations. Please see the revised version.

  • Authors should differentiate, whether they studied and reported the phenotype under water deficit stress or drought stress. Technically water deficit stress and drought stress are relevant but not the exactly same.

Response: Thank you for pointing this. We used 40 % Field Capacity and termed as water deficit stress. As per suggestion word drought is replaced by water deficit stress in the whole text. Please see the revised version.

  • Authors used two different formulas for hydrogen peroxide (H2O2) and H2O2, I will suggest to the authors please confirm it and used the correct one.

Response: Thank you for pointing this. There was a typo-mistake and now it is corrected. We have checked the whole manuscript and used one correct form (H2O2).

  • The authors used L-methionine (Met; 4mM) as a foliar treatment. On which bases they selected this dose, please provide references from the previous study if any

Response: Thank you for pointing this. We have already conducted some preliminary experiments by using different concentrations of L-methionine (0 mM, 0.05 mM, 0.15 mM, 0.4 mM, 1 mM, 2 mM, and 4 mM). On the bases of findings from these experiments, we only used 4 mM concentration in this study.

  • Similarly, two wheat verities “Galaxy-13 and Johar-16” were used in this study. Why only these two verities were used.

Response: Thank you for pointing this. In previous studies Galaxy-13 and Johar-16 were reported as drought tolerant and susceptible, respectively (Wasaya et al., 2021 https://doi.org/10.3390/su13094799). So, in this study we also used these two verities as drought-tolerant (Galaxy-13) and drought-susceptible (Johar-16).

  • Line 85: “Heathy seeds were selected based” there is a spelling mistake, please confirm.

Response: Thank you for pointing this. Spellings are corrected, please see the line 100.

  • Sterilized seeds were dried, but authors didn’t mention how they were dried.

Response: Thank you for pointing this. Sterilized seeds were dried at room temperature. Please see the line 103.

  • In some places, authors added space and some places not, e.g., (deep; 50 cm, and circumference; 30cm). I will suggest to them please use one format.

Response: Thank you for pointing this. Suggestion is incorporated, please see the revised manuscript.

  • Authors measured the water use efficiency (WUE) by applying a formula. Please provide detail formula.

Response: Thank you for pointing this. Formula for water use efficiency (WUE) is provided. Please see the line 146.

  • Line 189: Please confirm the sentence, “hundredd grain weight (HGW) were calculated with standard protocols”.

Response: Thank you for pointing this. Suggestion is incorporated, please see the line 215.

  • Line 199: P values should be in italic form.

Response: Thank you for nice suggestion and it is incorporated. Please see the revised manuscript.

  • Line 214: please confirm the units of chlorophyll.

 Response: Thank you for nice suggestion and it is incorporated.

  • Please improve the discussion part by comparing the previous research in other crop species.

Response: Thank you for nice suggestion and it is incorporated. Please see the revised manuscript.

Reviewer 2 Report

Authors are requested to see the comments mentioned in attached file and do the necessary corrections

Author Response

Dear Reviewer, Thank you for your careful reading of our manuscript and your helpful comments and questions. We have carefully revised this manuscript according to yours comments. A point-by-point reply to these comments is below:

Author should also add data pertaining to loss in global wheat production due to drought/water deficit stress.

Response: Thank you for pointing this. Suggestion is incorporated. Please see the lines 70-72.

Line 58: the number 3 should be in subscript.

Response: Thank you for nice suggestion and it is incorporated. Please see the line 72.

Line 64: Rephrase the sentence.

Response: Thank you for nice suggestion and it is incorporated. Please see the line 77-79.

Line 69: Why glyoxalase?? add some point related to it.

Response: Thank you for nice suggestion and it is incorporated. Please see the lines 82-84.

Line 105: Authors have directly mentioned single concentration of methionine used for experiment. What is the reason to choose this concentration? Is some preliminary trial has been conducted to support the view of single concentration of methionine (4mM) used in present experiment? or some past literature has been analyzed to decide this optimum concentration? Pl specify it in material and methods section.

Response: Thank you for pointing this. We have already conducted some preliminary experiments by using different concentrations of L-methionine (0 mM, 0.05 mM, 0.15 mM, 0.4 mM, 1 mM, 2 mM, and 4 mM). On the bases of findings from these experiments, we only used 4 mM concentration in this study.

Line 111: Stage of physiological maturity refers to time period taken from sowing till complete maturity of grain/ biomass in case of wheat; which usually take approximately 130 days-150 days or more depending upon the variety. Is it the short duration variety used in experiment as 55 DAS is quite shorter to achieve physiological maturity? Check it.

Response: Thank you for pointing this. There was a typo-mistake and it is corrected. Please see the line 132.

Line 123: Some experimental errors have been seen in writing part of manuscript. The readings of instrument IRGA should be usually taken at the time maximum PAR interception (from 11.00-2.00 in present location) due to maximum efficiency achieved at high light intensity. If moderate sunlight conditions are used, it is better to mention PAR (Photosynthetically active radiation) intensity in it.

Response: Thank you for pointing this. IRGA readings were taken according to (Wasaya et al., 2021; Long and Bernacchi, 2003) and day was full sunny with with a CO2 concentration of 400 μmol mol-1.

Line 129: Mention the wavelength used.

Response: Thank you for pointing this. Wavelengths are provided. Please see the lines 155-156.

In the Table 1, Use the same font and do not italicize the heading.

Response: Thank you for pointing this. There was a typo-mistake and now it is corrected. Please see the table 1.

In the Table 1. Check the font.

Response: Thank you for pointing this. There was a typo-mistake and now it is corrected. Please see the table 1.

Line 212: NGPP or NSPP?

Response: Thank you for pointing this. There was a typo-mistake (NGPP is correct form) and now it is corrected. Please see the line 246.

Line 227: Instead of pooled increase; mention per cent increase by methionine for stress and non-stress separately.

Response: Thank you for nice suggestion and it is incorporated. Please see the line 282.

Line 252: Is it Johar 16 or Ujala 16? please clarify.

Response: Thank you for pointing this. In this study we used two wheat verities (Galaxy-13 and Johar-16). There was a typo-mistake and now it is corrected. Please see the lines 327.

Line 257: correct the spelling improved.

Response: Thank you for pointing this. Suggestion is incorporated, please see the line 329.

Line 262: Is it Johar 16 or Ujala 16? please clarify

Response: Thank you for pointing this. In this study we used two wheat verities (Galaxy-13 and Johar-16). There was a typo-mistake and now it is corrected. Please see the lines 333 and 339.

Line 322: Generally, heat maps are used for transcriptomic analysis. Authors are requested to check it once, as these are to be given in main manuscript or as supplement file.

Response: Thank you for pointing this. We agreed with the reviewer that the heat maps normally used for transcriptomic data but Heat Maps are graphical representations of data that utilize color-coded systems. For better understanding of the data, we also used heatmap and we present these heatmaps in the main text.

Line 324: Discussion needs to be elaborated. Authors should discuss his findings and the reasons to support that finding.

Response: Thank you for nice suggestion. We have tried to improve the discussion part and we hope that now it will be acceptable.

Line 327: check the spelling of GLOBE.

Response: Thank you for pointing this. Spellings are corrected, please see the line 437.

In the conclusion: The manuscript research is on drought stress not on salt induced stress. If multiple stresses is faced by drought stressed seedlings then rephrase the sentence.

Response: Thank you for nice suggestion and it is incorporated. Please see the line 513.